# Low Prevalence of *Plasmodium falciparum* Histidine-Rich Protein 2 and 3 Gene Deletions—A Multiregional Study in Central and West Africa

**DOI:** 10.3390/pathogens12030455

**Published:** 2023-03-14

**Authors:** Tina Krueger, Moses Ikegbunam, Abel Lissom, Thaisa Lucas Sandri, Jacques Dollon Mbama Ntabi, Jean Claude Djontu, Marcel Tapsou Baina, Roméo Aimé Laclong Lontchi, Moustapha Maloum, Givina Zang Ella, Romuald Agonhossou, Romaric Akoton, Luc Djogbenou, Steffen Borrmann, Jana Held, Francine Ntoumi, Ayola Akim Adegnika, Peter Gottfried Kremsner, Andrea Kreidenweiss

**Affiliations:** 1Institute of Tropical Medicine, University Hospital Tübingen, 72074 Tübingen, Germany; 2Departement of Pharmaceutical Microbiology and Biotechnology, Nnamdi Azikiwe University, Awka 420007, Nigeria; 3Molecular Research Foundation for Students and Scientist, Nnamdi Azikiwe University, Awka 420007, Nigeria; 4Fondation Congolaise pour la Recherche Médicale, Brazzaville, The Republic of the Congo; 5Department of Zoology, Faculty of Science, University of Bamenda, Bambili P.O. Box 39, Cameroon; 6Synovo GmbH, Paul-Ehrlich-Strasse 15, 72076 Tübingen, Germany; 7Centre de Recherches Médicales de Lambaréné, Lambaréné P.O. Box 242, Gabon; 8Fondation Pour la Recherche Scientifique (FORS), ISBA, Cotonou, Bénin; 9Tropical Infectious Diseases Research Centre (TIDRC), University of Abomey-Calavi, Cotonou, Benin; 10German Center for Infection Research (DZIF), partner site Tübingen, 72074 Tübingen, Germany

**Keywords:** *Plasmodium falciparum*, histidine-rich protein 2 and 3, *pfhrp2*/*pfhrp3* deletions, rapid diagnostic test, Central Africa, West Africa

## Abstract

*Plasmodium falciparum* parasites carrying deletions of histidine-rich protein 2 and 3 genes, *pfhrp2* and *pfhrp3*, respectively, are likely to escape detection via HRP2-based rapid diagnostic tests (RDTs) and, consequently, treatment, posing a major risk to both the health of the infected individual and malaria control efforts. This study assessed the frequency of *pfhrp2-* and *pfhrp3*-deleted strains at four different study sites in Central Africa (number of samples analyzed: Gabon *N* = 534 and the Republic of Congo *N* = 917) and West Africa (number of samples analyzed: Nigeria *N* = 466 and Benin *N* = 120) using a highly sensitive multiplex qPCR. We found low prevalences for *pfhrp2* (1%, 0%, 0.03% and 0) and *pfhrp3* single deletions (0%, 0%, 0.03% and 0%) at all study sites (Gabon, the Republic of Congo, Nigeria and Benin, respectively). Double-deleted *P. falciparum* were only found in Nigeria in 1.6% of all internally controlled samples. The results of this pilot investigation do not point towards a high risk for false-negative RDT results due to *pfhrp2*/*pfhrp3* deletions in Central and West African regions. However, as this scenario can change rapidly, continuous monitoring is essential to ensure that RDTs remain a suitable tool for the malaria diagnostic strategy.

## 1. Introduction

Malaria represents a major health burden mainly affecting Sub-Saharan Africa. The majority of the more than 247 million cases and 619,000 deaths per year occur in Sub-Saharan Africa [1]. Malaria is caused by *Plasmodium* spp. parasites, with *Plasmodium falciparum* being the deadliest and most prevalent species in Africa. In order to reduce the burden and spread of the disease, prevention, therapy and early and accurate diagnosis are crucial. The primary tool for malaria diagnostics comprises microscopic detection of stained parasites and rapid diagnostic tests (RDT). The latter is of particular importance in rural African areas where microscopy services are difficult to implement due to limited availability of equipment, electricity and trained personnel. The majority of RDTs in use contain monoclonal antibodies against *P. falciparum* histidine-rich protein 2 (HRP2) that also cross-react to a certain level with the structurally related *P. falciparum* histidine-rich protein 3 (HRP3) [2,3]. These proteins are *P. falciparum*-specific and are expressed in all stages of the asexual life cycle from subtelomeric genes on chromosome 7 for *pfhrp2* and chromosome 13 for *pfhrp3*. The function of HRP2 is not fully understood, but it has been proposed to be involved in heme detoxification, modulation of infected red blood cells or host immune response [4,5,6].

Parasites that lack *pfhrp2* and optionally *pfhrp3* can cause false-negative results in HRP2-based RDTs. In settings with a well-established test-and-treat strategy, *pfhrp2/pfhrp3*-deleted strains could escape diagnosis and treatment and thus could eventually be positively selected from the parasite population [7]. First deletions have been found between 2003 and 2008 in South America with prevalences of 41% for *pfhrp2* and 70% for *pfhrp3* [8]. In the last decade, many more deleted *P. falciparum* strains were found all across the globe in Asia, the Middle East and Africa [1]. Based on the malaria threat map provided by the World Health Organization (WHO) to track studies, deleted strains have been identified in 37 of 44 countries under investigation with largely varying prevalences [9]. It is hypothesized that deletions in *pfhrp2* and *pfhrp3* genes occur naturally in the population due to their subtelomeric location but the drivers of selection are not well understood. Some studies found lower parasitemia in sub-sets of patients with *pfhrp2-* and *pfhrp3*-deleted parasites, suggesting a fitness cost that has been supported in experimental competition assays using gene edited parasites [10,11,12]. Mathematical models have identified low transmission and high treatment rates after HRP2-based diagnosis as key factors for the spread of deleted strains [7].

The WHO has recognized *pfhrp2* and *pfhrp3* gene deletions as one of the major threats to malaria elimination and calls for urgent action on monitoring the prevalence of these deletions [13]. Alternative non-HRP2-based RDTs are currently not widely available and often inferior to HRP2-based RDTs in regard to sensitivity or stability [14,15]. The WHO recommends changing the national diagnostic strategy to non-HRP2-based diagnostics only if a prevalence threshold of 5% for *pfhrp2/pfhrp3* gene deletions causing false-negative RDT results is exceeded. So far, this has happened in Eritrea, Ethiopia and Djibouti, leading to a switch to non-HRP2-based routine malaria diagnostics [1]. The prevalence of gene deletions differs widely, both within and between malaria-endemic countries, and many countries have never conducted epidemiologic studies on the prevalence of *pfhrp2/pfhrp3*-deleted parasites. To date, there have not been any peer-reviewed data published from Benin or the Republic of Congo. For Benin, a first study on deletions has recently been presented at the American Society of Tropical Medicine and Hygiene (ASTMH) annual meeting that found 21 *pfhrp2* deletions in 471 *P. falciparum* isolates (4.5%) [16]. Both countries have neighboring countries where the presence of deletions has been reported, such as the Democratic Republic of Congo with a prevalence of 6.4% or Nigeria with *pfhrp2* deletions in 16% of the analyzed samples [17,18].

Most studies utilize classic PCR methods followed by gel electrophoresis and visual analysis to identify gene deletions [17,18,19,20]. Although one or more single-copy genes are used as reference, these approaches are prone to misclassification due to the qualitative read-outs. A recent comparison of digital droplet PCR (ddPCR) and nested PCR (nPCR) demonstrated an increased risk of overestimation of deletions in studies relying on nPCR [21]. We recently developed a highly sensitive multiplex quantitative PCR (4-plex qPCR) targeting four genes within one participant sample [22]. The 4-plex qPCR is suited for large-scale screening as it detects and confirms *P. falciparum* infection and identifies *pfhrp2* and *pfhrp3* deletions against a single-copy control gene [22]. The internally controlled single-tube approach makes it a rapid and specific tool, which facilitates the identification of regional hotspots in need of adapted control strategies. Recently, two additional multiplex qPCR approaches have been developed and, so far, have been applied to first sample sets from African countries [23,24]. The aim of this study was to apply the 4-plex qPCR to large cohorts to provide a first assessment of the prevalence of *pfhrp2/pfhrp3* deletions in different regions of Central and West Africa.

## 2. Materials and Methods

This retrospective, cross-sectional, epidemiological study was performed on samples collected in specific areas from Gabon, Congo, Nigeria and Benin. Figure 1 gives a geographic overview of the involved study sites and Table 1 details the sampling and study population characteristics. In Gabon, samples collected from 2019 to 2020 from individuals older than two years and living in semi-urban Lambaréné and rural surroundings were included. The study design has been previously described [22]. Whole blood was collected into ethylenediamine tetra-acetic acid (EDTA) tubes. In Congo, individuals older than one year and living in the southern parts of the country (rural Goma Tse-Tse district; urban Brazzaville) were enrolled. Whole blood was sampled from March to September 2021 covering both the dry and the rainy seasons [25]. In Nigeria, whole blood was sampled from 2018 to 2019 from individuals in hospital settings located in semi-urban areas in Anambra, the state of southeastern Nigeria. In Benin, individuals of at least one year of age and living in rural areas of the Kpomasse-Tori Bossito health district in southern Benin were enrolled. Capillary blood was collected between June and October 2019 [26] and stored on filter paper as dried blood spots. 

Demographic characteristics were assessed from questionnaires. Individuals were considered symptomatic if they had a fever of over 37.5 °C. For parasitological assessment, fresh blood was collected and directly used for RDT and/or microscopy. RDTs were performed using the Paracheck Pf RDT (detects HRP2 only) in Gabon and the Malaria P.f./Pan Antigen kit, Cypress Diagnostics, in the Republic of Congo. This was conducted at the time of sample collection. In Nigeria, RDTs with the Standard Diagnostics Bioline Malaria Ag P.f Test kit detecting only HRP2 were performed for a sub-set of samples but, for consistency reasons, were not considered in this analysis. In Benin, no RDTs were performed. In addition, microscopy was performed in Gabon, Congo and Benin to detect and quantify *P. falciparum* as described previously [25,27]. Microscopic reading was performed by two independent microscopists. For the samples originating from Nigeria, no microscopy was performed.

For Gabon, Congo and Nigeria, DNA was extracted from whole EDTA blood using the QIAamp DNA mini Kit (Qiagen, Hilden, Germany) or the Quick-DNA Miniprep kit (Zymo-Research, Freiburg, Germany) according to the manufacturer’s instructions. In Benin, DNA was extracted from dried blood spots using the Chelex protocol as described previously [28]. The DNA was used in the 4-plex qPCR based on the amplification of the four target genes: *cytochrome b* (*pfcytb*), *ß tubulin* (*pfbtub*), *pfhrp2* and *pfhrp3*. The qPCR was performed at the Institute of Tropical Medicine, Tuebingen, Germany, for the samples from Gabon, Nigeria and Benin, and at the Centre de Recherches sur les Maladies Infectieuses (CeRMI), Brazzaville, Republic of Congo, for the samples from Congo. The 4-plex qPCR protocol was based on our previous publication with the following modifications [22]: *pfhrp2* and *pfhrp3* primers and probe sequences were modified to increase mismatches to the oligo-binding region of the respective homologous gene (oligonucleotide sequences are given in Table A1), and the *pfbtub* probe now comprises an internal TAO quencher in order to decrease the background signal of the quality control assay. The 4-plex qPCR reaction was performed at the following final concentrations: 1× TaqMan Multiplex Master Mix (ThermoFisher Scientific, Waltham, MA, USA), 400 nM for *pfcytb*, *pfhrp2* and *pfhrp3* forward and reverse primers each and 600 nM for *pfbtub* primers. The final probe concentrations were 300 nM for *pfbtub*, 75 nM for *pfhrp2* and 150 nM for *pfhrp3* and *pfcytb*. The 4-plex qPCR was performed in 384-well or 96-well plates on a LightCycler 480 I or II (Roche Diagnostics, Basel, Switzerland) with a final volume of 10 µL including 3 µL template or 20 µL including 6 µL template DNA. For each instrument, color compensation was performed and applied to the respective experiments before analysis. Cycling conditions included an initial activation step at 95 °C for 20 s, 45 cycles at 95 °C for 3 s, followed by 62 °C for 150 s and a final cooling step at 40 °C. The adapted protocol was characterized and showed a comparable limit of detection of 0.06 parasites/µL for *pfcytb* and 0.6 parasites/µL for *pfbtub*, *pfhrp2* and *pfhrp3*.

The qPCR was validated in each setting with five different controls: DNA extracted from the *P. falciparum* laboratory strains 3D7 (*pfhrp2*+/*pfhrp3*+), Dd2 (*pfhrp2*−/*pfhrp3*+), HB3 (*pfhrp2*+/*pfhrp3*−), uninfected whole blood as non-template control and water. Samples were measured in duplicates or triplicates with at least one positive control and one negative control on each plate. Discordant duplicates were repeated. Genotyping of the *P. falciparum* chloroquine resistance transporter (*pfcrt*) was performed using a multiplex qPCR assay as described previously [29].

LightCycler 480 II (Roche Diagnostics, Basel, Switzerland, Version 1.5.1.62) software was used and quantification cycles (C_q_) were obtained via absolute quantification using the second derivative maximum method or fit point method with the threshold above the negative controls [30,31]. The result was interpreted as positive for the respective gene if at least two out of two or three replicates showed amplification (C_q_ < 40). A sample was considered negative for the respective gene if there was no amplification in any of the duplicates or no more than one in the triplicates. Plates with no amplification of the positive control or a signal in the negative control were excluded from further analysis and repeated. Confirmed *P. falciparum* positive samples (positive for *pfcytb*) were considered deleted if they were positive for the single-copy control gene *pfbtub* and negative for *pfhrp2* and/or *pfhrp3* in at least two independent experiments.

Demographic characteristics were expressed via absolute numbers and percentages for categorical variables. For the agreement between the different diagnostic methods, Cohen’s kappa or Fleiss kappa was calculated as appropriate [32]. The strength of agreement can be interpreted as fair, moderate or substantial for kappa values ranging from 0.2–0.4, 0.4–0.6 and 0.6–0.8, respectively [33]. The difference of C_q_ values between false-positive and true-positive RDTs was evaluated using the Mann–Whitney U-test. The statistical analysis and graphical presentations were conducted in R (V4.0.2) (R Core Team, Vienna, Austria).

## 3. Results

### 3.1. Baseline Characteristics

A total number of 2037 participants were included in 4 different study regions in Gabon (*N* = 534), Congo (*N* = 917), Nigeria (*N* = 466) and Benin (*N* = 120).

The baseline characteristics of the study participants are displayed in Table 1. The median age of the study population ranged from 9 years in Benin to 16 years in Gabon (no information on age was available for the Nigerian cohort). A detailed description of the demographics, symptoms and diagnostic test results of 773 out of the 917 individuals from the Congolese study population can be found elsewhere [25]. The Congolese, Nigerian and Beninese cohorts comprised both symptomatic and asymptomatic individuals. For Gabon, no data on the symptom status were available. 

### 3.2. Malaria Diagnosis Outcome

For diagnosis of *P. falciparum* infections, thick blood smear (TBS) microscopy, malaria rapid diagnostic tests (RDT) and 4-plex qPCR (*pfcytb* for *P. falciparum* detection) were performed and the results are presented in Table 2.

The concordance of positive test results of samples, which were systematically tested with all three diagnostic methods, was analyzed (Figure 2). As expected, there was moderate to substantial agreement between microscopy of Giemsa-stained TBS and RDT—two methods that have a similar limit of detection. Considering microscopy as the gold standard, there were 8/95 (8%) and 2/329 (0.6%) cases of false-negative HRP2-RDT results in Gabon and Congo, respectively, resulting in an RDT sensitivity of 92% and 99%, respectively. PCR-based methods, especially qPCR, are more sensitive than microscopy and RDT, and can detect sub-microscopic infections. Compared to the highly sensitive 4-plex qPCR (*pfcytb*), there were 91/271 (34%) and 186/643 (29%) false-negative samples by HRP2-RDT in Gabon and Congo, respectively. For samples with a false-negative RDT result, the Cq values for *P. falciparum*/*pfcytb* were significantly higher, indicating a lower parasitemia compared to RDT true-positive samples (*p* < 0.001). This indicates that low parasitemia could account for many of the false-negative RDT outcomes (Figure A2).

### 3.3. Molecular pfhrp2 and pfhrp3 Deletion Detection

Amongst the samples confirmed for *P. falciparum* infection and for sufficient DNA template for single-copy gene amplification, thus double-positive samples for *pfcytb* and *pfbtub*, no *pfhrp2* and/or *pfhrp3* deletion was found in Congo and Benin (Table 3). In Gabon, two *pfhrp2*-deleted *P. falciparum* parasites were found. Double-deleted parasites were only detected in Nigeria. The qPCR curves for each of the single- and double-deleted *P. falciparum* samples can be found in Figure A3 and A4, respectively.

Some authors have used multiplex qPCR to identify polyclonal infections of deleted and non-deleted strains based on the C_q_ difference between the *pfhrp2*/*pfhrp3* and the single-copy reference gene *pfbtub* [24]. We did not find any indication for a large extent of multiclonal infections with deleted and non-deleted strains in Gabon, Congo or Benin, but a tendency towards a bimodal distribution of C_q_ differences in the samples from Nigeria (see Figure A1).

The *pfhrp2* single deletions in Gabon originated from a 22-year-old male and a female individual of unknown age who tested negative in both microscopy and HRP2-RDT (see Table 4). In Nigeria, one single *pfhrp2* deletion with a positive RDT test result and one single *pfhrp3* deletion without available RDT data were found. The double deletions that were found in Nigeria originated from the Nnewi (*n* = 2), Onitsha (*n* = 2) and Awka (*n* = 1) regions. All samples with detected *pfhrp2*/*pfhrp3* deletions underwent further genotyping for chloroquine resistance. Four of the deleted strains from Nigeria were chloroquine resistant (2× SVMNT and 2× CVIET haplotype). For the remaining three and the samples from Gabon, genotyping was not possible due to limited sample material.

## 4. Discussion

The use of HRP2-based RDTs as a main tool for malaria diagnosis and case management in Sub-Saharan Africa is threatened by *P. falciparum* parasites carrying *pfhrp2*/*pfhrp3* gene deletions. Here, we present data from cohorts covering a wide geographical region in Central and West Africa of high malaria endemicity, including the southern region of Congo as a country without available respective data so far, as well as specific areas in Gabon, Nigeria and Benin. To our knowledge, this is one of the first large cross-regional molecular screenings for *pfhrp2*/*pfhrp3* deletions of the *P. falciparum* population using a multiplex qPCR.

The prevalence of *pfhrp2* gene deletions was low in all of the included study locations, with the highest prevalences of 6/316 (1.9%) found in the cohort from southern Nigeria and 1% in Gabon, which are in line with our previous data from Gabon [22]. The presence of false-negative RDTs in our study is more likely to be explained by a lower load of parasites that can still be detected by the more sensitive qPCR reference test. The countries of all our study sites are highly endemic for *P. falciparum* with incidences of 200–400 annual cases per 1000 at risk compared to 40–55 in Eritrea or Ethiopia, where many deletions have been found [1]. This might be one reason for the low frequency of identified deletions in this study, as low malaria prevalence besides a stringent test-and-treat strategy was stated as one of the most important factors for selection of deletions [7].

In a study with febrile children in Nigeria from 2019, double-deleted parasites were found in 6% of the samples analyzed with molecular methods [18]. This corresponds to a total prevalence of 1.3%, which is very similar to our results in asymptomatic participants. A study on travelers returning from Nigeria to Australia between 2011 and 2015 found 13% single-*pfhrp2* deletions but no double deletions by PCR [20]. For Benin and Congo, this is the first *pfhrp2/pfhrp3* deletion study and no mutated parasites have been detected. Studies from neighboring countries, using classical or nested PCR combined with qualitative visual agarose gel analysis for the read-out, found variable prevalences for *pfhrp2* deletions of up to 22% in a nationwide study in the Democratic Republic of Congo in 2013–2014 and up to 30% in Ghana in 2015 [3,17]. We could not find any deletions in Brazzaville, while the authors reported deletions in 20% of the isolates from Kinshasa, which is in close geographical proximity to Brazzaville, only separated by the Congo River. Besides the geographical distance, those samples were collected from a different study population, during a different study period, and analyzed with conventional PCR to identify deletions that might be more prone to misclassification due to the non-quantitative read-out [21].

Our study is strengthened by the use of a reliable, high-throughput method to sensitively detect *P. falciparum* infections and analyze deletions. With four reactions taking place in a single tube, we can internally control the presence of a sufficient amount of DNA with *pfbtub* as a single-copy reference gene and, consequently, decrease the risk of falsely classifying parasites as deleted. Moreover, the 4-plex qPCR has been established and used in the Republic of Congo, extending the molecular toolbox in an endemic setting for potential future surveillance activities. This study allowed us to further validate our 4-plex qPCR as a highly specific tool to identify *pfhrp2/pfhrp3* deletions. One of the main limitations of the present study is its retrospective nature and the incompleteness of the metadata, especially of RDT results. This does not allow us to draw conclusions about the extent of HRP2-RDT failure. Other limitations include the sampling that was limited to a specific area per country and not representative of the whole population, and the procedures that were only partially harmonized between the study centers. Our results inform on the regional deletion frequencies of the *P. falciparum* population circulating in the respective study groups; extrapolation to the whole country and to the general population should be conducted with caution. Regional or temporal hotspots within these countries with higher deletion prevalences could have been missed. Future studies should be prospective and representative in design, use harmonized procedures, be conducted over a larger time period covering different seasons and consider more geographically diverse locations within the countries concerned or even national coverage to account for regional patterns of deleted strains [17].

Studies on *pfhrp2*/*pfhrp3* deletions can be prone to several sources of bias. Deletions can be overestimated as a consequence of genetic variability in the oligo-binding domains in these subtelomeric *P. falciparum* genes [23,34]. However, the low number of deletions we have found in the high number of *P. falciparum*-positive samples shows that our experimental setup is well suited to detect *pfhrp2* and *pfhrp3* in field samples. Furthermore, there is the risk of underestimation of deletions due to genetic variability or partial deletions of exons [23,34,35]. Results from several studies targeting both exons and flanking regions suggest a deletion pattern of the whole gene rather than partial deletions [17,20,36,37]. Moreover, in a high-malaria-transmission setting, the chance to detect *pfhrp2*-deleted parasites decreases because polyclonal infections with *pfhrp2*-deleted and wild-type parasites are more likely to occur and can mask the deleted strain. Thus, in areas or seasons of high transmission, the true prevalence of *pfhrp2* deletions is likely to be underestimated. However, the distribution of C_q_ values did not hint towards a high level of polyclonal infections with deleted strains.

The WHO suggests changing the national diagnostic strategy to non-HRP2-based diagnostics if a threshold of 5% of *pfhrp2*/*pfhrp3* deletions causing false-negative RDTs is reached [13]. An accurate methodology to assess the prevalence is critical, as overestimation would lead to a complex, laborious and costly switch to non-HRP2-based RDTs that have limited availability, sensitivity and stability [38]. Due to the large variety in the design and methodologies of studies, the WHO provided a standard surveillance protocol [39]. A recent comparison with ddPCR pointed out the risk of nested PCR falsely classifying deletions [21]. A systematic assessment of the currently available molecular tools for deletion analysis, including qPCR, ddPCR, nPCR and conventional PCR, is needed to compare the diagnostic accuracies [21,23,24].

Many aspects of emerging *pfhrp2*/*pfhrp3* deletions remain unclear, such as the clinical consequences, transmission dynamics, the interaction between the deleted parasite, host and vector and, most importantly, the drivers of selection. Modeling suggests low transmission with high coverage of HRP2-based testing and treatment as key factors for selection. Although *pfhrp2* single-deleted and *pfhrp2*/*pfhrp3* double-deleted strains have recently been shown to carry a fitness cost in in vitro growth competition assays, deleted strains have become dominant in several independent contexts, even in locations without a widespread use of HRP2-based RDTs [10]. Future research is needed to unravel the factors that drive potential selection of deleted strains and to reveal more information about the virulence and transmission of these strains.

In conclusion, our data do not currently point towards an increased risk of high false-negative HRP2-RDTs due to *pfhrp2* deletions in the respective study locations in Gabon, Republic of Congo, Nigeria and Benin. However, nationwide regular monitoring procedures should be implemented in order to enable timely detection of potential deletions and thus the implementation of corrective measures, if indicated.

## Figures and Tables

**Figure 1 pathogens-12-00455-f001:**
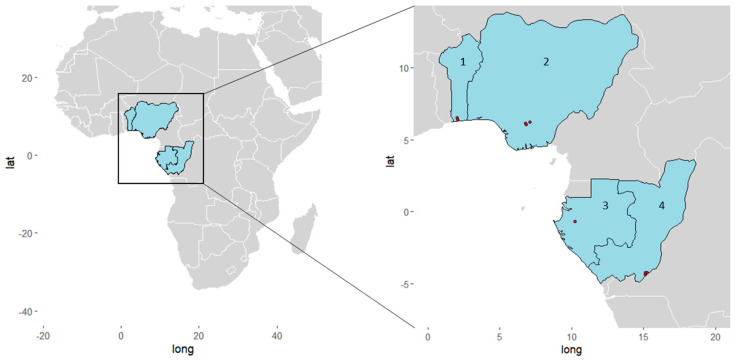
Geographical overview of the study sites in West and Central Africa. Red dots represent sampling sites (1) Benin: Kpomasse, Ouidah; (2) Nigeria: Awka, Nnewi, Onitsha; (3) Gabon: Lambaréné; (4) Republic of Congo: Goma Tsé-Tsé District, Brazzaville. The map was created with the ggplot2 package V3.4.0 in R Version 4.2.2.

**Figure 2 pathogens-12-00455-f002:**
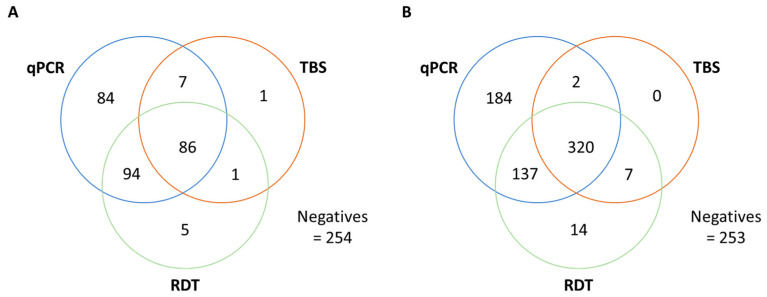
Venn diagram of concordant *P. falciparum* diagnostics. Concordance of positive test results between the 4-plex qPCR (*P. falciparum* detection via *pfcytb*), RDT and TBS microscopy in (**A**) Gabon and (**B**) the Republic of Congo. Only samples with available results for all three diagnostic tests (qPCR, TBS, RDT) were included. RDT: Malaria rapid diagnostic test, TBS: Thick blood smear microscopy.

**Table 1 pathogens-12-00455-t001:** Demographic characteristics of the study populations.

	Gabon	Congo	Nigeria	Benin
Total *N*	534	917	466	120
Age in years:				
Median (IQR)	16 (6–25)	15 (8–38)	*	9 (5–13)
Age cohorts:				
0–6 years, *n* (%)	77 (14%)	169 (18%)	-	45 (38%)
7–18 years, *n* (%)	91 (17%)	334 (36%)	-	49 (41%)
18+ years, *n* (%)	134 (25%)	412 (45%)	-	26 (22%)
Missing data	232 (43%)	2 (0.2%)	466 (100%)	-
Sex:				
Female	258 (48%)	514 (56%)		75 (63%)
Male	154 (29%)	403 (44%)		45 (38%)
Missing data:	122 (23%)	-	466 (100%)	-
Symptomatic:	*	73/914 (8%)	299 (64%)	9 (8%)
Sampling period:	2019–2020	2021	2018–2019	2019
Number of study sites:				
1	Lambaréné 534 (100%)	Goma Tsé-Tsé District 573 (62%)	Awka 200 (43%)	Kpomasse 68 (57%)
2	-	Brazzaville 344 (38%)	Nnewi 167 (36%)	Ouidah 52 (43%)
3	-	-	Onitsha 99 (21%)	-
Measurements:				
TBS done, *n* (%)	534 (100%)	917 (100%)	not done	120 (100%)
RDT done, *n* (%)	532 (99.6%)	917 (100%)	not available	not done
RDT brand:	Paracheck Pf RDT	Malaria P.f./Pan Antigen kit, Cypress Diagnostics	-	-

* Data are missing. IQR: Interquartile range TBS: Thick blood smear microscopy, RDT: Malaria rapid diagnostic test.

**Table 2 pathogens-12-00455-t002:** *P. falciparum* diagnostic outcomes.

	Gabon	Congo	Nigeria	Benin
Total *N*	534	917	466	120
TBS positive, *n* (%)	96 (18%)	329 (36%)	-	120 (100%)
Median parasitemia [p/µL]	475 (145–1849)	530 (147–5514)	-	2240 (1008–6900)
RDT positive, *n* (%)	186/532 * (35%)	478 (52%)	-	-
qPCR positive	273 (51%)	643 (70%)	379 (81%)	120 (100%)
False-negative RDTs vs TBS vs qPCR	8 91	2 186	- -	- -
Kappa				
RDT/TBS	0.50	0.67	-	-
RDT/qPCR	0.64	0.54	-	-
TBS/qPCR	0.33	0.36	-	-
RDT/TBS/qPCR	0.47	0.50	-	-

* Data are missing. qPCR positivity refers to *pfcytb* positivity.

**Table 3 pathogens-12-00455-t003:** Prevalence of *pfhrp2*, *pfhrp3* and *pfhrp2*/*pfhrp3* double deletions stratified by study site.

	Gabon	Republic of Congo	Nigeria	Benin
*pfcytb* and *pfbtub* positive, *n*	218	512	316	120
*pfhrp2*-deleted	2 (1%)	0 (0%)	1 (0.03%)	0 (0%)
*pfhrp3*-deleted	0 (0%)	0 (0%)	1 (0.03%)	0 (0%)
*pfhrp2-* and *pfhrp3*-deleted	0 (0%)	0 (0%)	5 (1.6%)	0 (0%)

*pfcytb* positivity indicates the presence of *P. falciparum* infection, while *pfbtub* as single-copy gene serves as an internal quality control for *pfhrp2*/*pfhrp3* deletion analysis.

**Table 4 pathogens-12-00455-t004:** Patient characteristics and diagnostic test results of the identified deletions.

ID	Country/ Subregion	TBS Result	HRP2-RDT Result	4plex qPCR				CQR
				*pfhrp2* C_q_	*pfhrp3* C_q_	*pfcytb* C_q_	*pfbtub* C_q_	*Haplotype*
1	Gabon/*	Neg	Neg	del	38.2	34.0	35.3	*
2	Gabon/*	Neg	Neg	del	37.1	31.7	34.5	*
3	Nigeria/Onitsha	*	Pos	del	37.4	34.6	35.0	CVIET
4	Nigeria/Nnewi	*	*	34.8	del	33.9	35.3	*
5	Nigeria/Nnewi	*	*	del	del	35.7	34.7	*
6	Nigeria/Nnewi	*	*	del	del	36.2	35.4	SVMNT
7	Nigeria/Onitsha	*	*	del	del	35.2	36.2	SVMNT
8	Nigeria/Onitsha	*	*	del	del	35.1	35.5	CVIET
9	Nigeria/Awka	*	*	del	del	31.3	37.0	*

* Data are missing; C_q_—cycle of quantification, Yrs—years, F—female, M—male, Neg—negative, Pos—positive, del—deleted, CQR—chloroquine resistance.

## Data Availability

Data are available upon request to the corresponding author.

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
