# Peer review of "Low Prevalence of *Plasmodium falciparum* Histidine-Rich Protein 2 and 3 Gene Deletions—A Multiregional Study in Central and West Africa"

_pathogens, 2023, doi:10.3390/pathogens12030455_

Round 1

Reviewer 1 Report

The authors evaluated the prevalence of pfhrp2/3-deleted isolates in more than 2,000 samples collected from four African countries between 2019 and 2021 using a multiplex qPCR method that they developed earlier. Pfhrp2/3 deleted parasites were not found in the Republic of Congo and Benin, while very few samples were found to be Pfhrp2/3 deleted in Gabon and Nigeria.

The background section is adequate. The methods are well described. The results are well presented and very clear. Discussion is pertinent.

MAJOR COMMENTS:

None

MINOR COMMENTS:

Line 41 “Sub-Saharan-Africa”: delete the hyphen between Saharan and Africa.

Line 41 “Plasmodium spp.”: Plasmodium in italics; spp. is not in italics.

Line 49 “HRP2 that also cross-bind to a certain level to…”: cross-react to a certain level with

Lines 56-57: pfhrp2/3-deleted strains could escape diagnosis and treatment

Line 61: World Health Organization (WHO)

Lines 76-77 “this has happened in Eritrea, Ethiopia and Djibouti…”: Please add reference citations.

Line 82 “ASTMH”: American Society of Tropical Medicine and Hygiene (ASTMH); poster or oral presentation?

Lines 83-84 “Both countries [Benin and the Republic of Congo] have neighbouring countries where a high prevalence of deletions has been reported”: Please make the statement clearer. Which neighboring countries? Please also add reference citations of the neighboring countries being referred to.

Lines 85-86 “Most studies utilize classic PCR methods…to identify gene deletions”: Please add some reference citations to support this statement.

Lines 89-90 “we recently developed a highly sensitive multiplex qPCR targeting four genes”: Please cite these four gene targets (given in line 135).

Lines 103-115: These ethical statements from four countries can be placed elsewhere. There are “institutional review board statement” and “informed consent statement” sections after the conclusions of the paper. The ethical statements do not have to be repeated in the main text (lines 103-115).

Line 119 “> 37.5°C”: Please specify whether the authors are referring to axillary or rectal (especially in children) temperature.

Lines 121-122 “Paracheck Pf RDT, Malaria P.f./Pan antigen kit”: Please specify whether these RDTs have additional target (pLDH). For the second RDT, I think that the readers will immediately understand that pLDH (Pan) is also used.

 Line 123 “SD Bioline”: There are several versions of SD Bioline RDT. Was the one used in Nigeria based on only HRP2 or was there also pLDH?

Line 130 “EDTA”: ethylenediaminotriacetic acid (EDTA)

Line 133-135: Please indicate in which laboratory (city, country) qPCR was performed.

Line 150 “p/µL”: Does “p” stand for parasites? Please clarify.

Line 156 “Pfcrt”: Plasmodium falciparum chloroquine resistance transporter (Pfcrt); “Pfcrt” in italics

Lines 158-160 “second derivative maximum method, fit point method”: Please provide reference citations.

Line 166 “cytb”: in italics

Lines 179-183, figure 1: In my opinion, this figure describes the study sites and is more appropriate in the materials and methods section.

Line 189 “both, symptomatic and asymptomatic”: delete the comma after “both”

Line 200 “there is good agreement”: was

Line 200 “good agreement”: It would be helpful if the authors can provide in the Materials and Methods section (or in Table 2 legend) the scales of kappa values so that readers can understand immediately that the values given in Table 2 correspond to “good” agreement. If the classification of kappa values is added, please also add the reference citation.

Line 202 “considering gold standard microscopy as the reference”: Considering microscopy as the gold standard

Lines 204-205 “PCR-based methods, especially qPCR, is more sensitive”: PCR-based methods…are more sensitive

Line 207 “there were 91/271 and 186/643 samples false-negative by HRP2-RDT”: there were 91/271 and 186/643 false-negative samples by HRP2-RDT

 Lines 208-210 “Cq values…were significantly lower…”: Please give the P-value. The statistical test(s) used for data analysis (other than kappa statistics) is/are not described in the Methods section (lines 170-171). Please add this information. Also please check the meaning of the statement. A false-negative RDT is expected to have low parasitemia as one of the possible causes. Low parasitemia should be reflected by high Cq values. True positive RDT is expected to be characterized by low Cq.

Line 229, Table 3: Data presented in this table are clearly described in the text. This table is not necessary.

Line 239 “…male and a female…that were tested negative”: who were tested negative

Line 243: delete one of the two periods after (n=1).

Line 247, Table 4: Two columns (age and sex) can be deleted from this table.

Line 311, “big variety”: large variety or wide variety

Line 354, Table A1: target genes in italics. The column for “supplier” is not necessary. This column can be deleted. The information that is more important is Tm values for each primer pair and the expected size in base pairs. These two pieces of information can be added and replace “supplier.”

Line 356: gene abbreviations in italics

Lines 359, 365, 368: in italics – P. falciparum, gene abbreviations

The references are in several formats. Please see the journal instructions and follow the journal style for references listed at the end of the manuscript.

REF 1 is incomplete.

REF 9: Please provide more information, including the web link.

REF 14: Please correct the article title.

REF 15 is incomplete: authors, journal, volume, pages.

REF 16: More information should be provided. Please check the journal instructions for conference proceedings.

REF 34: Please complete this reference.

Author Response

Dear Editor

Dear Reviewers

We appreciate your consideration and the possibility to submit a revised version of our manuscript titled “Low Prevalence of Plasmodium falciparum Histidine Rich Protein 2 and 3 Gene Deletions – A Multiregional Study in Central and West Africa”. Thank you for your valuable feedback that helped us to improve our manuscript in the current version. Based on your comments, we have revised our paper. Apart from editorial corrections, the main changes included a more detailed description of the study population and a more differentiated discussion of the introduced bias due to the study design and the study locations. Please find our point-by-point responses for each reviewer below. Clear, self-explanatory requests (mainly for editorial changes) have been implemented and are denoted with the check mark () symbol. All changes in the manuscript are marked via Track Changes. Line numbers (LL) refer to manuscript with track changes accepted.

Reviewer 1:

Comments and Suggestions for Authors

The authors evaluated the prevalence of pfhrp2/3-deleted isolates in more than 2,000 samples collected from four African countries between 2019 and 2021 using a multiplex qPCR method that they developed earlier. Pfhrp2/3 deleted parasites were not found in the Republic of Congo and Benin, while very few samples were found to be Pfhrp2/3 deleted in Gabon and Nigeria.

The background section is adequate. The methods are well described. The results are well presented and very clear. Discussion is pertinent.

MAJOR COMMENTS:

None

MINOR COMMENTS:

  • Line 41 “Sub-Saharan-Africa”: delete the hyphen between Saharan and Africa.
  • Line 41 “Plasmodium spp.”: Plasmodium in italics; spp. is not in italics.
  • Line 49 “HRP2 that also cross-bind to a certain level to…”: cross-react to a certain level with
  • Lines 56-57: pfhrp2/3-deleted strains could escape diagnosis and treatment
  • Line 61: World Health Organization (WHO)
  • Lines 76-77 “this has happened in Eritrea, Ethiopia and Djibouti…”: Please add reference citations.
    • The reference has been added.
  • Line 82 “ASTMH”: American Society of Tropical Medicine and Hygiene (ASTMH); poster or oral presentation?
    • We updated the sentence and introduced the abbreviation correctly. The reference is the abstract from the abstract book that is available online. It is not stated clearly, if the work as been presented in a poster or an oral presentation.
  • Lines 83-84 “Both countries [Benin and the Republic of Congo] have neighbouring countries where a high prevalence of deletions has been reported”: Please make the statement clearer. Which neighboring countries? Please also add reference citations of the neighboring countries being referred to.
    • This has been added.
  • Lines 85-86 “Most studies utilize classic PCR methods…to identify gene deletions”: Please add some reference citations to support this statement.
  • Lines 89-90 “we recently developed a highly sensitive multiplex qPCR targeting four genes”: Please cite these four gene targets (given in line 135).
  • Lines 103-115: These ethical statements from four countries can be placed elsewhere. There are “institutional review board statement” and “informed consent statement” sections after the conclusions of the paper. The ethical statements do not have to be repeated in the main text (lines 103-115).
    • The ethical statements have been moved to respective sections after the conclusion.
  • Line 119 “> 37.5°C”: Please specify whether the authors are referring to axillary or rectal (especially in children) temperature.
    • This classification refers to the axillary temperature.
  • Lines 121-122 “Paracheck Pf RDT, Malaria P.f./Pan antigen kit”: Please specify whether these RDTs have additional target (pLDH). For the second RDT, I think that the readers will immediately understand that pLDH (Pan) is also used.
    • The Paracheck Pf RDT is only based on HRP2. This piece of information has been added in the manuscript.
  • Line 123 “SD Bioline”: There are several versions of SD Bioline RDT. Was the one used in Nigeria based on only HRP2 or was there also pLDH?
    • The SD Bioline RDT in Nigeria was based only on HRP2. We have adapted the text passage to make it clearer.
  • Line 130 “EDTA”: ethylenediaminotriacetic acid (EDTA)
    • A more detailed descriptions of the study population has been added, where the abbraviation EDTA has been introduced (line 110)
  • Line 133-135: Please indicate in which laboratory (city, country) qPCR was performed.
    • The following sentence was added in line 148: “The qPCR was performed at the Institute of Tropical Medicine, Tuebingen, Germany for the samples from Gabon, Nigeria and Benin and at the Centre de Recherches sur les Maladies Infectieuses (CeRMI), Brazzaville, the Republic of Congo for the samples from Congo.”
  • Line 150 “p/µL”: Does “p” stand for parasites? Please clarify.
    • The text has been changed from p/µL to parasites/µL.
  • Line 156 “Pfcrt”: Plasmodium falciparum chloroquine resistance transporter (Pfcrt); “Pfcrt” in italics
  • Lines 158-160 “second derivative maximum method, fit point method”: Please provide reference citations.
    • References have been added.
  • Line 166 “cytb”: in italics
  • Lines 179-183, figure 1: In my opinion, this figure describes the study sites and is more appropriate in the materials and methods section.
    • The authors agree. Figure 1 and the respective text reference were moved to the method section.
  • Line 189 “both, symptomatic and asymptomatic”: delete the comma after “both”
  • Line 200 “there is good agreement”: was
  • Line 200 “good agreement”: It would be helpful if the authors can provide in the Materials and Methods section (or in Table 2 legend) the scales of kappa values so that readers can understand immediately that the values given in Table 2 correspond to “good” agreement. If the classification of kappa values is added, please also add the reference citation.
    • Thank you for pointing this out. The classification of kappa values according to Landis & Koch (1977) was added in the method section. In addition, good was replaced by the more appropriate terms moderate and substantial according to the kappa values and their interpretation.
  • Line 202 “considering gold standard microscopy as the reference”: Considering microscopy as the gold standard
  • Lines 204-205 “PCR-based methods, especially qPCR, is more sensitive”: PCR-based methods…are more sensitive
  • Line 207 “there were 91/271 and 186/643 samples false-negative by HRP2-RDT”: there were 91/271 and 186/643 false-negative samples by HRP2-RDT
  • Lines 208-210 “Cq values…were significantly lower…”: Please give the P-value. The statistical test(s) used for data analysis (other than kappa statistics) is/are not described in the Methods section (lines 170-171). Please add this information. Also please check the meaning of the statement. A false-negative RDT is expected to have low parasitemia as one of the possible causes. Low parasitemia should be reflected by high Cq values. True positive RDT is expected to be characterized by low Cq.
    • The difference of Cq values between false-positive and true-positive RDTs was evaluated using the Mann-Whitney-U-test with a p-value of 2.2e-16. These information were added in the manuscript (as p<.001).
    • There was a mistake in the statement. Indeed, the words higher and lower were mixed up. We have corrected this in the manuscript.
  • Line 229, Table 3: Data presented in this table are clearly described in the text. This table is not necessary.
    • We can understand your comment and agree, that a part of the information in the table and the text is redundant. However, we prefer to keep the table, as it directly summarizes the key results of our work. To make it more concise, we have shortened the respective paragraph in the text.
  • Line 239 “…male and a female…that were tested negative”: who were tested negative
  • Line 243: delete one of the two periods after (n=1).
  • Line 247, Table 4: Two columns (age and sex) can be deleted from this table.
    • The respective columns have been deleted.
  • Line 311, “big variety”: large variety or wide variety
  • Line 354, Table A1: target genes in italics. The column for “supplier” is not necessary. This column can be deleted. The information that is more important is Tm values for each primer pair and the expected size in base pairs. These two pieces of information can be added and replace “supplier.”
    • The table has been updated and the information on Tm and amplicon size added.
  • Line 356: gene abbreviations in italics
  • Lines 359, 365, 368: in italics – P. falciparum, gene abbreviations
  • The references are in several formats. Please see the journal instructions and follow the journal style for references listed at the end of the manuscript.
    • We agree and have updated the reference format.
  • REF 1 is incomplete.
  • REF 9: Please provide more information, including the web link.
  • REF 14: Please correct the article title.
  • REF 15 is incomplete: authors, journal, volume, pages.
  • REF 16: More information should be provided. Please check the journal instructions for conference proceedings.
  • REF 34: Please complete this reference.

Reviewer 2 Report

Authors present data on prevalence of pfhrp2/pfhrp3 deletions as detected by a multiplex-qPCR in four countries of Central and West African region. These surveys are important as the spread of Plasmodium falciparum populations harbouring these deletions may compromise the accuracy of Rapid Diagnostic of malaria and consequently the control or eradication strategies in endemic areas.

The study is straightforward and well-presented and only minor issues need to be addressed.

- Bibliographic references need to be revised and some of them completed (eg – what is reference 15? False-negative RDT results and implications of new reports of P. falciparum histidine-rich protein 2/3 gene deletions. 2017.)

- Introduction

Line 82 – “… recently been presented on the ASTMH Annual meeting

- Materials and Methods

A better description of study populations is needed. For instance, it is said that “Detailed description of the recruitment process and study population as well as geographical details of the study sites have been published elsewhere[22,23].” (lines 116-117) but references 22 and 23 relate only to Republic of Congo and Benin. I suppose that samples from Gabon result from the study referenced as 24 and still there is no reference for the Nigeria collection but all this information is a bit confused.

- Results

Lines 189-190 and Table 1 – It is stated that “… in Benin, only asymptomatic cases were included.” but 9(8%) of symptomatic individuals are presented in Table 1 for Benin. Please clarify.

Still about this information – in table 2, median parasitaemia is much higher in Benin that in Gabon or Congo, which is a bit puzzling as these individuals, although younger, are said to be all asymptomatic. Could the authors discuss a bit more or give more information on this result?

Lines 208-210 – I believe that there is a mistake in this sentence – “For samples with a false-negative RDT result the Cq values for P. falciparum/pfcytb were significantly HIGHER (NOT LOWER), which indicates a LOWER (NOT HIGHER) parasitemia compared to RDT true positive samples”, in order to agree with the next sentence and Figure A2. Please confirm.

Table 2 and Figure 2 – The prevalence of pfhrp2/3 deletions was found to be low but still there are 91 and 186 false negative RDTs in Gabon and Republic of Congo, respectively when compared to qPCR (33.5% and 28.9% of qPCR positives). According to the results, this would not be due to the presence of deleted parasites and possible reasons for this should also be discussed.

Further in this table, figures should be revised – summing up values presented at the Venn diagrams, totals are 271 qPCR positive in Gabon (not 273) and 643 (not 644) in the Republic of Congo.

Author Response

Dear Editor

Dear Reviewers

We appreciate your consideration and the possibility to submit a revised version of our manuscript titled “Low Prevalence of Plasmodium falciparum Histidine Rich Protein 2 and 3 Gene Deletions – A Multiregional Study in Central and West Africa”. Thank you for your valuable feedback that helped us to improve our manuscript in the current version. Based on your comments, we have revised our paper. Apart from editorial corrections, the main changes included a more detailed description of the study population and a more differentiated discussion of the introduced bias due to the study design and the study locations. Please find our point-by-point responses for each reviewer below. Clear, self-explanatory requests (mainly for editorial changes) have been implemented and are denoted with the check mark () symbol. All changes in the manuscript are marked via Track Changes. Line numbers (LL) refer to manuscript with track changes accepted.

Reviewer 2:

Authors present data on prevalence of pfhrp2/pfhrp3 deletions as detected by a multiplex-qPCR in four countries of Central and West African region. These surveys are important as the spread of Plasmodium falciparum populations harbouring these deletions may compromise the accuracy of Rapid Diagnostic of malaria and consequently the control or eradication strategies in endemic areas.

The study is straightforward and well-presented and only minor issues need to be addressed.

  • Bibliographic referencesneed to be revised and some of them completed (eg – what is reference 15? False-negative RDT results and implications of new reports of P. falciparum histidine-rich protein 2/3 gene deletions. 2017.) 
    • The references have been revised and corrected.

- Introduction

  • Line 82 – “… recently been presented on the ASTMH Annual meeting
    •  

- Materials and Methods

A better description of study populations is needed. For instance, it is said that “Detailed description of the recruitment process and study population as well as geographical details of the study sites have been published elsewhere[22,23].” (lines 116-117) but references 22 and 23 relate only to Republic of Congo and Benin. I suppose that samples from Gabon result from the study referenced as 24 and still there is no reference for the Nigeria collection but all this information is a bit confused.

  • We have added details on the study population. Indeed, the given references provide more information on the study design and population from Congo and Benin. For Gabon, the samples have been specifically collected for this study. The samples from Nigeria have not yet been described in another publication so far.

- Results

  • Lines 189-190 and Table 1 – It is stated that “… in Benin, only asymptomatic cases were included.” but 9(8%) of symptomatic individuals are presented in Table 1 for Benin. Please clarify.
    • Thank you for pointing this out. This is indeed a mistake, as both asymptomatic and symptomatic patients were included in Benin. We have corrected the respective statement in the manuscript.
  • Still about this information – in table 2, median parasitaemia is much higher in Benin that in Gabon or Congo, which is a bit puzzling as these individuals, although younger, are said to be all asymptomatic. Could the authors discuss a bit more or give more information on this result?
    • This should also be resolved by the explanation above. The samples in Benin originated from symptomatic and asymptomatic individuals. The sentence has been corrected.
  • Lines 208-210 – I believe that there is a mistake in this sentence – “For samples with a false-negative RDT result the Cq values for P. falciparum/pfcytb were significantly HIGHER (NOT LOWER), which indicates a LOWER (NOT HIGHER) parasitemia compared to RDT true positive samples”, in order to agree with the next sentence and Figure A2. Please confirm.
    • Thank you for pointing this out. Indeed, there was a mistake and the words lower and higher were mixed up. We have corrected the sentence in the manuscript.
  • Table 2 and Figure 2 – The prevalence of pfhrp2/3 deletions was found to be low but still there are 91 and 186 false negative RDTs in Gabon and Republic of Congo, respectively when compared to qPCR (33.5% and 28.9% of qPCR positives). According to the results, this would not be due to the presence of deleted parasites and possible reasons for this should also be discussed.
    • This is likely to be explained by lower parasitemia as the reference test qPCR has a higher sensitivity than RDTs. The statement has been corrected in the results section and an additional sentence has been included in the discussion.
  • Further in this table, figures should be revised – summing up values presented at the Venn diagrams, totals are 271 qPCR positive in Gabon (not 273) and 643 (not 644) in the Republic of Congo.
    • Thank you for this comment. For the Republic of Congo, the total number for the total qPCR positives was corrected from 644 to 643 in Table 2. For Gabon, the discrepancy between the number in the Venn diagram and the table arises from 2 cases with missing RDT (see Table 1 - RDT done is 532/534 or Table 2) as the Venn diagram includes only cases with complete data on each of the three different diagnostic test (also mentioned in the figure description). The sentence has been rephrased for better understanding.

Reviewer 3 Report

Pathogens-2239406-peer-review-comments-v1

Overall Comments :

The study deals with an important issue concerning malaria diagnosis and assessment of HRP2/3 deletions. The study was retrospective and utilized samples collected from four study sites, two each in Central Africa (Gabon and Republic of Congo) and West Africa (Nigeria and Benin) using a highly sensitive multiplex-qPCR (evaluated and published by the authors previously). The study design was multicentric and cross-sectional and the results showed a low prevalence for pfhrp2- (1%, 0%, 28 0.03% and 0) and pfhrp3-single deletions (0%, 0%, 0.03% and 0%) at all study sites (Gabon, the Republic of Congo, Nigeria and Benin, respectively). Double-deleted P. falciparum were only found in Nigeria in 1.6% of all internally controlled samples.

Overall, the paper is well written and covers an important topic essential for individual patient treatment as well as the overall goal of malaria elimination. Systematic studies on the PFHRP2 and 3 gene deletions are recommended by the WHO also to arrive at meaningful conclusions regarding the policy changes in malaria diagnostic tools. However, I have some concerns regarding this study and request clarifications and appropriate revisions from the authors on the following points :

·        I am not comfortable denoting this study as a multicentric study for estimating the prevalence of Histidine Rich Protein 2 and 3 Gene Deletions since a multicentric epidemiological study per se is essentially expected to follow a common protocol for at least the study design, sample size, sampling techniques, study period and the methodology employed in processing and analysis of the samples, which is apparently not the case in this particular study.

·        Since the conclusion of this study is expected to have a bearing on the country’s diagnosis policy, it would be interesting to know whether the sample size was estimated based on the known prevalence in the study areas(at least those for which data is available) or following epidemiologically relevant sampling methodology?

·        Can the study sites be denoted as representative of the four areas i.e., Gabon, Republic of Congo, Nigeria and Benin? The demographic profile (references 22 and 23) shows that the samples were collected from specific areas i.e., in southern districts of Brazzaville in The Republic of the Congo during 2021 for one study and from the rural localities in the Ouidah–Kpomasse–Tori Bossito (OKT) health district in Southern Benin from June to October 2019 for another study. There are differences in the study design too between these two studies. This fact has not been brought out appropriately in the materials and methods, results and discussion.

·        Since blood was collected from all age groups as whole blood or stored on filter paper in the frame of ongoing studies, it would be relevant to know whether the variation in results between the whole blood and DBS samples was ruled out.

·        It is unclear how the study participants signed informed consent to participate in this study since it was retrospective and originally the samples collected were intended to address different objectives(references 22 and 23).

·        Is it possible for the authors to estimate the positive and negative predictive values of the novel 4plex qPCR for Pf HRP 2/3 deletions from the available data?  It might be useful to know this technique's sensitivity, specificity and predictive value and present it in this paper for future use.

·        Although it is good to see that the authors have already recommended that prospective systematic studies should be conducted in future for monitoring the prevalence of HRP 2/ 3 deletions following the WHO recommended surveillance methodology, I do not see enough merit in the methodology or justification in the discussion to generalize the findings of this study to the entire Gabon and Republic of Congo as well as Nigeria and Benin.  This might lead to an unnecessary false sense of security in the said areas. Please discuss the bias that is likely to be introduced in the study due to the difference in the study areas , sampling, study design etc. and how likely it is to influence the outcome of this study.

Given the above concerns and limitations of this study, it is recommended that appropriate changes are made in the paper to address these for publication of this paper.

Specific comments :

·        Please change the font of the tables/figure titles to bold.

·        Since this is an epidemiological study, it is desirable to add details about the specific study areas in this paper for a better understanding of the readers, researchers and policymakers of these specific countries.

·        Italicize P.falciparum  in the entire text

·        Line 297-291: Please justify the results appropriately given this limitation and consider adding a few lines in the results section also for clarity to readers.

·        Considering restricting the results and conclusions of this study to the specific study areas only due instead of generalizing them. It might be appropriate to emphasize more the diagnostic value of the 4plex qPCR for Pf HRP 2/3 deletions rather than the prevalence in the study areas. Please rethink and consider revisions to address the technical concerns.

·        Consider changing the title of the study given the concerns expressed for calling this a multicentric study.

Author Response

Dear Editor

Dear Reviewers

We appreciate your consideration and the possibility to submit a revised version of our manuscript titled “Low Prevalence of Plasmodium falciparum Histidine Rich Protein 2 and 3 Gene Deletions – A Multiregional Study in Central and West Africa”. Thank you for your valuable feedback that helped us to improve our manuscript in the current version. Based on your comments, we have revised our paper. Apart from editorial corrections, the main changes included a more detailed description of the study population and a more differentiated discussion of the introduced bias due to the study design and the study locations. Please find our point-by-point responses for each reviewer below. Clear, self-explanatory requests (mainly for editorial changes) have been implemented and are denoted with the check mark () symbol. All changes in the manuscript are marked via Track Changes. Line numbers (LL) refer to manuscript with track changes accepted.

Reviewer 3:

Overall Comments :

The study deals with an important issue concerning malaria diagnosis and assessment of HRP2/3 deletions. The study was retrospective and utilized samples collected from four study sites, two each in Central Africa (Gabon and Republic of Congo) and West Africa (Nigeria and Benin) using a highly sensitive multiplex-qPCR (evaluated and published by the authors previously). The study design was multicentric and cross-sectional and the results showed a low prevalence for pfhrp2- (1%, 0%, 28 0.03% and 0) and pfhrp3-single deletions (0%, 0%, 0.03% and 0%) at all study sites (Gabon, the Republic of Congo, Nigeria and Benin, respectively). Double-deleted P. falciparum were only found in Nigeria in 1.6% of all internally controlled samples.

Overall, the paper is well written and covers an important topic essential for individual patient treatment as well as the overall goal of malaria elimination. Systematic studies on the PFHRP2 and 3 gene deletions are recommended by the WHO also to arrive at meaningful conclusions regarding the policy changes in malaria diagnostic tools. However, I have some concerns regarding this study and request clarifications and appropriate revisions from the authors on the following points :

  • I am not comfortable denoting this study as a multicentric study for estimating the prevalence of Histidine Rich Protein 2 and 3 Gene Deletions since a multicentric epidemiological study per se is essentially expected to follow a common protocol for at least the study design, sample size, sampling techniques, study period and the methodology employed in processing and analysis of the samples, which is apparently not the case in this particular study.
    • We fully agree to your comment. The term multicentric has been replaced by multiregional throughout the manuscript.
  • Since the conclusion of this study is expected to have a bearing on the country’s diagnosis policy, it would be interesting to know whether the sample size was estimated based on the known prevalence in the study areas (at least those for which data is available) or following epidemiologically relevant sampling methodology?
    • When the Gabon study was designed, no information on the prevalence of hrp2 and/or hrp3 deleted P. falciparum parasites were available for Gabon. Sample size was based on the assumption of 2% hrp2 deletions in the region. We assumed that if deletions occurred at higher rates, this large proportion of false negative RDT tests would not remain unnoticed by physicians at the study site.
    • As defined in the methods, LL 104, this was a retrospective, cross-sectional, epidemiological study. Strictly speaking, the sampling was not representative for the whole respective country. We modified the manuscript at various locations and limit the findings more to respective sampling areas. Thus, we write in the abstract we write: LL31: The results of this pilot investigation do not indicate towards a high risk for false-negative RDT results due to pfhrp2/pfhrp3 deletions in Central and West African regions. Introduction, LL100: The aim of this study was to apply the 4plex qPCR to large cohorts to provide a first assessment of the prevalence of pfhrp2/pfhrp3-deletions in different regions of Central and West Africa. Discussion, LL261: Here, we present data from cohorts covering a wide geographical region in Central and West Africa with high malaria endemicity, including the southern region of Congo as a country without available respective data so far, as well as specific areas in Gabon, Nigeria, and Benin. And LL302: Other limitations include the sampling that was limited to a specific area per country and not representative for the whole population, and the procedures that were only partially harmonized between the study centers. Our results inform on the regional deletion frequencies of the P. falciparum population circulating in the respective study groups; extrapolation to the whole country and to the general population should be done with caution. Regional or temporal hotspots within these countries with higher deletion prevalence could have been missed. Future studies should be prospective and representative in design, use harmonized procedures, be conducted over a larger time periode covering different seasons, and consider more geographically diverse locations within the countries concerned or even national coverage to account for regional pat-terns of deleted strains[17].
  • Can the study sites be denoted as representative of the four areas i.e., Gabon, Republic of Congo, Nigeria and Benin? The demographic profile (references 22 and 23) shows that the samples were collected from specific areas i.e., in southern districts of Brazzaville in The Republic of the Congo during 2021 for one study and from the rural localities in the Ouidah–Kpomasse–Tori Bossito (OKT) health district in Southern Benin from June to October 2019 for another study. There are differences in the study design too between these two studies. This fact has not been brought out appropriately in the materials and methods, results and discussion.
    • In terms of malaria endemicity, the sampling areas in Gabon and in Congo largely represent the level of endemicity in the respective country and is comparable to those in the sampling areas of Benin and Nigeria. However, in the latter countries endemicities in the northern parts of the countries are different (lower). We account now for the limits of the study and add a word of caution concerning the extrapolation of the results to the whole countries at various instances throughout the manuscript (see details above).
  • Since blood was collected from all age groups as whole blood or stored on filter paper in the frame of ongoing studies, it would be relevant to know whether the variation in results between the whole blood and DBS samples was ruled out.
  • For DNA extraction and 4plex qPCR whole blood samples were used in Gabon, Congo and Nigeria and dried blood spots (DBS) in Benin. In our paper published earlier on the establishment and performance evaluation of the 4plex qPCR we used both whole blood and DBS (see REF22 (original REF19), Kreidenweiss et al.). What we have seen is that the sensitivity for the detection of P. falciparum (pfcytb) in DBS samples is somewhat lower than in whole blood. However, this has little impact on the determination of pfhrp2/pfhrp3 prevalence due to the two internal quality controls of the 4plex PCR, the way of sample collection hardly influences the result of the detection of pfhrp2/pfhrp3 deletions. Only samples that are P. falciparum = pfcytb positive and have sufficient genetic material for the detection of single copy genes = pftub positive are used for the pfhrp2/pfhrp3 analysis.
  • It is unclear how the study participants signed informed consent to participate in this study since it was retrospective and originally the samples collected were intended to address different objectives(references 22 and 23).
  • All studies included here are malaria studies approved by the respective ethical committees and signed informed consent. The Gabon study was designed specifically for hrp2/hrp3 deletion assessment. All other studies include sampling to address P. falciparum infections and malaria with various objectives including P. falciparum diagnosis and assessments.
  • Is it possible for the authors to estimate the positive and negative predictive values of the novel 4plex qPCR for Pf HRP 2/3 deletions from the available data? It might be useful to know this technique's sensitivity, specificity and predictive value and present it in this paper for future use.
    • PPV and NPV are measures relevant for diagnostic or screening tools used for patient care. The 4plex qPCR is a molecular assay to investigate the P. falciparum genetic make-up that is here pfhrp2 and pfhrp3. There is no reference test for hrp2/hrp3 deletion detection. Beyond this, the detection of something that is not there – is conceptionally very challenging anyway. Thus, PPV and NPV are not appropriate parameters for hrp2/hrp3 molecular deletion detection. However, we could determine PPV and NPV for the diagnosis of Pf infection by pfcytb. So far, we did not do comparator qPCRs for the diagnosis of Pf as this was not the primary aim of the 4plex qPCR.
  • tAlthough it is good to see that the authors have already recommended that prospective systematic studies should be conducted in future for monitoring the prevalence of HRP 2/ 3 deletions following the WHO recommended surveillance methodology, I do not see enough merit in the methodology or justification in the discussion to generalize the findings of this study to the entire Gabon and Republic of Congo as well as Nigeria and Benin. This might lead to an unnecessary false sense of security in the said areas. Please discuss the bias that is likely to be introduced in the study due to the difference in the study areas , sampling, study design etc. and how likely it is to influence the outcome of this study. Given the above concerns and limitations of this study, it is recommended that appropriate changes are made in the paper to address these for publication of this paper.
    • We agree to this finding and have added several sentences throughout the manuscript. For details, please see our response to your first point.

Specific comments :

  • Please change the font of the tables/figure titles to bold.
    • The table and figure titles have been formatted according to the template that the journal provides. There, only the term “figure” or “table” with the respective number is bold.
    • Since this is an epidemiological study, it is desirable to add details about the specific study areas in this paper for a better understanding of the readers, researchers and policymakers of these specific countries.
    • A more detailed description of the study sites and populations has been added in the methods and can also be found in Table one. We added if sampling was done in rural or urban areas, added information regarding malaria endemicity, year/periode of sampling, etc..
    • Italicize P.falciparum in the entire text
    • Line 297-291: Please justify the results appropriately given this limitation and consider adding a few lines in the results section also for clarity to readers.
      • If we understand it correctly, this comment concerns the “limitation” non-quantitative read-out. However, this refers to the cited study from the Democratic Republic of Congo. In our study, we have used the quantitative 4plex qPCR, which we also highlight in the following sentence. We have replace “these” by “those” to make more clear that we are referring to the cited study.
    • Considering restricting the results and conclusions of this study to the specific study areas only due instead of generalizing them. It might be appropriate to emphasize more the diagnostic value of the 4plex qPCR for Pf HRP 2/3 deletions rather than the prevalence in the study areas. Please rethink and consider revisions to address the technical concerns.
    • This has been adapted in the method, results and conclusion section. Moreover, we have pointed it out in the discussion, that the results from this study should not be generalized to a nationwide prevalence estimate. For details, please see our response to your first point.
    • Consider changing the title of the study given the concerns expressed for calling this a multicentric study.
      • We agree. The term multicentric has been replaced by multiregional throughout the manuscript.

Round 2

Reviewer 3 Report

Dear Authors,

The changes and responses are well accepted. I feel the paper has been adequately revised for publication.

Wish you all the best !